# New Perspectives in Radiological and Radiopharmaceutical Hybrid Imaging in Progressive Supranuclear Palsy: A Systematic Review

**DOI:** 10.3390/cells12242776

**Published:** 2023-12-06

**Authors:** Joachim Strobel, Hans-Peter Müller, Albert C. Ludolph, Ambros J. Beer, Nico Sollmann, Jan Kassubek

**Affiliations:** 1Department of Nuclear Medicine, University Hospital Ulm, 89081 Ulm, Germany; ambros.beer@uniklinik-ulm.de; 2Department of Neurology, University Hospital Ulm, 89081 Ulm, Germany; hans-peter.mueller@uni-ulm.de (H.-P.M.); albert.ludolph@rku.de (A.C.L.); jan.kassubek@uni-ulm.de (J.K.); 3German Center for Neurodegenerative Diseases (DZNE), Ulm University, 89081 Ulm, Germany; 4Department of Diagnostic and Interventional Radiology, University Hospital Ulm, 89081 Ulm, Germany; nico.sollmann@tum.de; 5Department of Diagnostic and Interventional Neuroradiology, School of Medicine, Klinikum rechts der Isar, Technical University of Munich, 81675 Munich, Germany; 6TUM-Neuroimaging Center, Klinikum rechts der Isar, Technical University of Munich, 81675 Munich, Germany

**Keywords:** progressive supranuclear palsy, neurodegenerative diseases, radiological biomarkers, radiopharmaceutical biomarkers, diagnosis

## Abstract

Progressive supranuclear palsy (PSP) is a neurodegenerative disease characterized by four-repeat tau deposition in various cell types and anatomical regions, and can manifest as several clinical phenotypes, including the most common phenotype, Richardson’s syndrome. The limited availability of biomarkers for PSP relates to the overlap of clinical features with other neurodegenerative disorders, but identification of a growing number of biomarkers from imaging is underway. One way to increase the reliability of imaging biomarkers is to combine different modalities for multimodal imaging. This review aimed to provide an overview of the current state of PSP hybrid imaging by combinations of positron emission tomography (PET) and magnetic resonance imaging (MRI). Specifically, combined PET and MRI studies in PSP highlight the potential of [18F]AV-1451 to detect tau, but also the challenge in differentiating PSP from other neurodegenerative diseases. Studies over the last years showed a reduced synaptic density in [11C]UCB-J PET, linked [11C]PK11195 and [18F]AV-1451 markers to disease progression, and suggested the potential role of [18F]RO948 PET for identifying tau pathology in subcortical regions. The integration of quantitative global and regional gray matter analysis by MRI may further guide the assessment of reduced cortical thickness or volume alterations, and diffusion MRI could provide insight into microstructural changes and structural connectivity in PSP. Challenges in radiopharmaceutical biomarkers and hybrid imaging require further research targeting markers for comprehensive PSP diagnosis.

## 1. Introduction

Among the Parkinsonian syndromes, progressive supranuclear palsy (PSP) is a rare but severe form of Parkinsonism characterized by vertical supranuclear gaze palsy, postural instability, and cognitive impairment [1]. Specifically, PSP belongs to the four-repeat (4R) tauopathies, characterized by brain parenchymal accumulation of a specific isoform of the protein tau [2,3,4].

In the clinical context, the diagnosis of PSP may still be challenging, although the International Parkinson and Movement Disorder Society Clinical Diagnostic Criteria (MDS-PSP) capture the multifaceted phenotypical presentations of PSP by the application of four functional domains (i.e., ocular motor dysfunction, postural instability, akinesia, and cognitive dysfunction), which have been defined as clinical key features of PSP [5]. Based on molecular pathology, PSP as a 4R tau proteinopathy has various clinical subtypes, primarily determined by diverse neuroanatomical distribution patterns of the underlying tau pathology [6]. Neuronal loss and gliosis occur in the cortical and subcortical core areas. Additionally, there is the deposition of pathological tau protein in neurons, oligodendrocytes, and astrocytes. The tau pathology observed in neurons is primarily characterized by large globular neurofibrillary tangles, while oligodendroglial deposits are denoted as “coiled bodies”, and the tau pathology in astrocytes is identified by “tufted astrocytes”, also known as astrocytic tangles [3]. The distribution patterns of pathologies differ by subtype, with prevailing subcortical pathology, particularly in basal ganglia such as the globus pallidus, nucleus subthalamicus, and brainstem, including also the nucleus subthalamicus, substantia nigra, colliculi, locus coeruleus, and raphe nuclei [4]. The lower olives and dentate nucleus in the cerebellum are also affected, albeit to a lesser extent.

Corticobasal degeneration (CBD) is a significant differential diagnosis to PSP, due to them both being 4R tauopathies [7]. Specifically, CBD exhibits similar tau pathology as compared to PSP, affecting neurons, oligodendroglia, and astroglia. Pre-tangles in nerve cells are the most prominent neuropathology of tau in the cerebral cortex, basal ganglia, and brainstem. Astrocytic plaques are another component, appearing mainly in the cerebral cortex. In comparison to the “tufted astrocytes” found in PSP, the astrocytes in this case contain deposits in the distal processes, giving rise to a plaque-like structure. The oligodendroglial pathology is reminiscent of that in PSP, characterized by “coiled bodies”. Other characteristics include swollen neurons, neuronal loss, and surface-level laminar spongiosis, particularly in cortical areas like the frontal and cingulate cortex [8]. The astroglial pathology represents the initial lesion, while during early stages, neuronal pathology becomes dominant, whereas in advanced disease stages, glial pathology outweighs involvement of astroglia. Clinical symptoms include gradual rigidity, clumsy limb movements, and cortical signs such as alien or anarchic limb phenomenon, dystonic postures, and gait disruption [7].

Therefore, an early and accurate diagnosis of PSP is crucial for best patient care and clinical management [9], particularly in the light of the debilitating effects of the disease on patients and their families. Of note, PSP has a rapid progression rate and substantially affects a patient’s quality of life, entailing severe disability and mortality [10,11]. In addition, the considerable rate of misdiagnoses regarding PSP increases the burden on healthcare resources and affects the allocation of appropriate treatment options [12]. While there are no disease-modifying treatments available for PSP to date, management can be offered in the sense of primarily symptomatic and supportive care [1,13]. Hence, there is an urgent need for the development of disease-modifying therapies to improve patient outcomes and reduce the socio-economic burden of the disease [4].

The development of diagnostic biomarkers for PSP diagnosis is essential on the way to facilitate early diagnosis and accurate differentiation between Parkinsonian syndromes [14]. Neuroimaging provides the opportunity to identify neuroanatomical correlates of the clinical presentation and disease process of PSP in vivo, and several imaging techniques have shown potential as methods to derive diagnostic biomarkers [15,16,17,18]. As part of ongoing research for the diagnosis and monitoring of PSP, it is important to emphasize the development of innovative imaging techniques in combination with biochemical biomarkers. By combining positron emission tomography (PET) and magnetic resonance imaging (MRI), the PET/MRI hybrid imaging method stands out as a powerful tool for the evaluation of neurodegenerative diseases. It allows for the simultaneous assessment of structural, functional, and molecular aspects of the brain. This approach provides valuable insights into the pathological processes associated with neurodegenerative disorders, enabling earlier and more accurate diagnoses, as well as a comprehensive understanding of disease progression. In essence, the combination of PET and MRI may allow for a comprehensive assessment of PSP-related changes across the brain [15,19].

Against this background, this review aimed to provide an overview of the current state of the role of PET/MRI hybrid imaging in the assessment of PSP, following the review by Whitwell and colleagues that has been published together with the MDS-PSP clinical criteria in 2017 [5,15]. In addition, PET/MRI hybrid imaging is discussed in terms of the diagnostic potential in comparison with other techniques, and a perspective is provided on future directions for the development of novel diagnostic biomarkers (with a focus on radiopharmaceuticals).

## 2. Methods

### 2.1. Literature Search Strategy

A literature review was conducted with the Preferred Reporting Items for Systematic Reviews and Meta-Analyses (PRISMA) guidelines [20]. During the systematic search conducted in June 2023, articles were collected from PubMed (https://www.ncbi.nlm.nih.gov/pubmed/ (accessed on 31 August 2023)), which were published after the publication by Whitwell et al. [15].

The search queries and keywords were as follows: (a) “progressive supranuclear palsy” AND “positron emission tomography” AND “magnetic resonance imaging”; (b) “progressive supranuclear palsy” AND “positron emission tomography” AND “MRI”; (c) “progressive supranuclear palsy” AND “PET” AND “magnetic resonance imaging”; (d) “progressive supranuclear palsy” AND “PET” AND “MRI”. In total, this search yielded 42 database entries. These records were carefully checked for the following criteria: (a) the studies had to be published in a peer-reviewed journal in English language; (b) only human studies with in vivo cranial MRI were considered; (c) reviews and case reports (less than at least 3 contributing PSP patients) were excluded; (d) publications only concerning study planning (thus not reporting original findings) were excluded.

### 2.2. Identified Articles

Finally, 21 original research articles were considered for the present review, all covering combined PET/MRI studies or studies that performed both PET and MRI shortly after each other and used respective multi-modal data in the same study (Figure 1). Given the lack of studies making use of MRI beyond coregistration/normalization purposes, according to initial title and abstract screening, a focus was set on radiopharmacy and the PET component of the respective included studies. Given the exposed role of PET imaging and radiopharmaceuticals in this review, the first part of the results (Section 3.1) comprehensively summarizes methodological aspects of PET imaging, together with radiopharmaceutical developments relevant to the study of neurodegenerative diseases in general and PSP in particular. The second part of the results (Section 3.2) focuses on the findings of the 21 identified research articles.

## 3. Results

### 3.1. Methodological Overview for PET

Imaging by PET is based on the injection of radiolabeled tracers that usually bind to specific targets in the brain, mainly allowing for the visualization of changes in metabolism, blood flow, and receptor density [21]. In recent years, considerable advancements in neuroimaging techniques and developments of diverse radiotracers have enabled researchers to visualize tau in tauopathies [22]. The radioligands are typically labeled with radioisotopes such as carbon-11 [11C] or fluorine-18 [18F] that have relatively short half-lives, making them optimal for clinical and experimental imaging [23].

The widely recognized tracer [18F]fluorodeoxyglucose (F18-FDG), which has been firmly established s many decades, has been the subject of numerous studies [24,25,26]. Studies have revealed distinct metabolic patterns in regional glucose metabolism that mirror regional alterations in neural network activity in conditions such as idiopathic Parkinson’s Disease (IPD), multiple system atrophy (MSA), PSP, and CBD [27,28]. The utilization of [18F]FDG-PET is currently a topic of ongoing debate, as it has the potential to enhance diagnostic precision and assist in differentiation among these disorders [25,29,30,31,32].

#### 3.1.1. First-Generation Tau Tracers

As most first-generation tau PET tracers, [18F]FDDNP was first developed to image the paired helical filament and amyloid-beta (Aβ) plaques found in Alzheimer’s disease (AD), demonstrating non-specific binding to tau aggregates [33]. More recent work conducted with this tracer showed that there is an increased [18F]FDDNP binding in the midbrain, subthalamic regions, and cerebellar white matter of PSP patients [22]. Subcortical and cortical tracer uptake also increased with disease severity [34]. Nevertheless, a lack of specificity of this tracer raises concerns regarding its use in imaging PSP-related tau pathology, given its affinity for Aβ [35,36]. Specifically, while aiming to utilize this tracer for tau imaging, the elevated signal observed in the lateral temporal regions of dementia patients, compared to those without dementia, was attributable to amyloid-beta rather than tau [37]. This result has been supported by the concurrent decrease in cerebrospinal fluid (CSF) Aβ levels [38].

Among the first generation of tau PET tracers, [18F]AV-1451 (flortaucipir) demonstrated highly selective binding to tau over Aβ and obtained approval by the U.S. Food and Drug Administration (FDA) for diagnostic procedures [39,40]. However, there are mixed results concerning whether in vivo [18F]AV-1451 uptake correlates with the neuropathological distribution of tau characteristics for PSP [41]. Specifically, PSP patients exhibited increased [18F]AV-1451 retention in the midbrain and left dentate nucleus, distinguishing them from controls, CBD patients, and other frontotemporal lobe dementia patients [42,43]. In addition, CBD patients demonstrated elevated [18F]AV-1451 uptake bilaterally in the premotor and motor cortices compared to controls, as well as in the globus pallidus compared to both PSP patients and controls [44]. Notably, [18F]AV-1451 exhibits a high affinity for monoamine oxidase-B, highlighting a concern of off-target binding associated with this tracer [45].

The big choice of different [18F]THK derivatives offers distinct benefits and limitations. The initial PET study demonstrated that [18F]THK-523 showed higher retention in white matter (WM) compared to gray matter (GM) in healthy controls (HC) [46,47]. However, patients with AD exhibited elevated binding of [18F]THK-523 in various regions of the neocortex and hippocampus, which correlated with the distribution of neurofibrillary tangles (NFT) in post-mortem brain samples [48]. However, there was no relevant retention of [18F]THK-523 in non-AD tauopathies such as PSP, which hindered its potential clinical use [48]. To overcome the limitations of [18F]THK-523, derivates such as [18F]THK-5117 and [18F]THK-5317 were developed [49]. Initial studies primarily utilized [18F]THK-5317 and [18F]THK-5117, and both demonstrated promising results for imaging atypical Parkinsonism (AP) tau in vitro and in vivo [50,51]. Further, [18F]THK-5317 demonstrated improved clinical usefulness in distinguishing patients with CBD and PSP from those with AD [51]. However, compared with [18F]AV-1451, its sensitivity was slightly lower [52]. As an improvement, [18F]THK-5351 was developed to address the high WM binding observed in [18F]THK-5117 and [18F]THK-5317 [53,54]. It demonstrated lower non-specific binding to WM, higher affinity for AD tau, and higher selectivity for tau over Aβ [55]. In this regard, PET studies with [18F]THK-5351 indicated better GM/WM ratio, faster kinetics, and higher binding in regions susceptible to PSP und AD pathology compared to other derivatives of the THK spectrum and [18F]AV1451 [55,56,57]. While [18F]THK-5351 displayed some off-target binding in certain regions, it still served well for the assessment of non-AD tauopathies [53].

Another tracer of the first generation is [11C]PBB3 (Figure 2), which demonstrated rapid uptake and binding to NFT in mice and showed promising results in differentiating AD patients from HC with an increased PET signal in the medial and lateral temporal cortices as well as the frontal cortex [58]. In patients with PSP, the binding of [11C]PBB3 was prominently observed in frontoparietal WM, parietal GM, globus pallidus, subthalamic nucleus, red nucleus, and cerebellar dentate nucleus [59,60]. This finding corresponds to the recently defined distribution patterns of PSP-tau inclusions in the brain [3]. However, limitations such as lipophilic metabolites, off-target binding, and metabolic instability have restricted its use to date [61].

#### 3.1.2. Second-Generation Tau Tracers

Studies of [18F]PM-PBB3 (or [18F]-APN-1607) demonstrated a promising ability to capture and visualize the progressive uptake of the tracer in specific regions where tau pathology is deposited [62]. Initial reports of [18F]PM-PBB3 PET in humans did not show off-target binding in the basal ganglia, the thalamus, or in the monoamino-oxidase (MAO) system, but exhibited distinct binding in the choroid plexus [63]. Because of the off-target binding in the midbrain and basal ganglia, it represented a promising tracer especially for 4R tauopathies, as reflected by several publications [64,65,66]. Furthermore, in a study in PSP using [18F]PM-PBB3, there was an uptake in the midbrain, subthalamic nucleus, and cerebellar dentate nucleus [67]. Overall, [18F]PM-PBB3 appears to be a promising tracer for imaging tau pathologies.

Another second-generation tracer is [18F]PI-2620 [68,69]. This particular tracer exhibits rapid clearance from the brain and a strong binding affinity to tau aggregates, which corresponds to the progressive stages of tau pathology according to the Braak staging system in AD [70]. Importantly, [18F]PI-2620 does not show any non-specific binding to MAO-A or MAO-B, substantia nigra, basal ganglia, or choroid plexus, distinguishing it from several other candidate tracers [69,71]. Several studies have reported a consistent binding pattern of tau tracers in PSP within the midbrain, globus pallidus, and subthalamic nucleus [43,72,73,74]. Brendel et al. conducted a study that demonstrated the usefulness of [18F]PI-2620 in improving the accuracy of diagnosing PSP [68,75]. Their findings suggested that this tracer effectively differentiated PSP from alpha-synucleinopathies (such as PD), AD, and controls [22,75].

Studies on [18F]MK-6240 showed favorable brain kinetics and homogeneous distribution without WM labeling [76]. Initial research revealed high affinity to NFT in AD [77]. The tracer exhibited stable profiles with no off-target binding in non-target cortical regions or the basal ganglia, but showed off-target binding in the substantia nigra, striatum, meninges, ethmoid sinus, and clivus bone [77,78,79]. The binding pattern of [18F]MK-6240 correlated with NFT accumulation and Braak staging in Aβ+ patients without dementia and AD patients [78]. Compared with [18F]flortaucipir, the tracer showed similar features capable of binding in the regions that are prone to tau deposition in AD, but with higher standardized uptake value (SUV) ranges [80]. Additionally, [18F]RO948, another relatively new tracer, has emerged as a promising radiopharmaceutical substance for detecting tau pathology in subcortical regions [81]. It demonstrated a distinct binding pattern in the globus pallidus and substantia nigra, making it valuable for differentiating PSP from other neurodegenerative disorders [82]. While [18F]flortaucipir and [18F]PI-2620 have been used to visualize tau pathology in cortical regions, [18F]RO948 was found to be more suitable for detecting tau pathology in subcortical regions, which may be highly relevant in PSP [83].

In addition, there are other next-generation tau tracers that are currently less investigated and not widely available, particularly regarding 4R tauopathies [54]. One candidate is [18F]CBD-2115, which has been studied for in vitro investigations [84].

#### 3.1.3. Recent Development of Tracers

Recent approaches that have investigated the role of synaptic loss in PSP and CBD aimed to better understand tauopathies by mapping synaptic density based on the affinity for a presynaptic vesicle glycoprotein (SV2A) [85,86]. One of the radiopharmaceuticals is [11C]UCB-J. Holland et al. were able to demonstrate a significant correlation between synaptic loss and disease severity in PSP and amyloid-negative CBD [87]. These findings suggest that [11C]UCB-J could serve as a marker of disease and progression, offering potential utility in clinical trials [88,89].

An intriguing tracer is [11C]PK11195, a ligand specifically binding to the translocator protein (TSPO) [90]. Specifically, TSPO undergoes an upregulation process on the surface of activated microglia, facilitating the visualization of neuroinflammation within the living brain using PET imaging [91]. Neuroinflammation encompasses complex cellular and biochemical responses that transpire within the central nervous system (CNS) in response to various triggers [92]. The working hypothesis suggests that neuroinflammation could represent a central role in the initiation of various forms of dementia and neurodegenerative disorders like PSP, involving the activation of microglial cells (i.e., specialized immune cells residing within the CNS) [93].

### 3.2. General Characteristics of Identified Literature

This review identified 21 articles published between 2017 and 2023, which were identified based on PRISMA guidelines and extracted after application of the inclusion and exclusion criteria (Table 1, Column 1, Study number (SN) 1-21) [42,74,87,94,95,96,97,98,99,100,101,102,103,104,105,106,107,108,109,110,111].

Of note, the majority of studies used MRI merely for the purposes of spatial normalization, co-registration of MRI and PET data, and/or anatomical reference for segmentations of target structures, sometimes supplemented by volumetric assessments (Table 1, SN 1, 2, 3, 6, 7, 9, 10, 11, 13, 14, 15, 17, 18) [87,95,96,99,100,103,104,105,107,108,109,110,111]. Hence, relevant MRI acquisitions were restricted to anatomical sequences such as T1-weighted (T1w) or proton-density-weighted (PDw) imaging (Table 1, SN 1, 2, 3, 6, 7, 9, 10, 11, 13, 14, 15, 17, 18) [87,95,96,99,100,103,104,105,107,108,109,110,111]. Few studies made use of other techniques beyond those anatomical sequences, including diffusion-weighted imaging (DWI), functional MRI (fMRI), and magnetic resonance spectroscopy (MRS) (Table 1, SN 5, 8, 12, 16, 19, 20) [42,94,97,98,101,106]. Nearly all studies exclusively used 3-Tesla MRI systems (Table 1, SN 1-21 except 4 and 15) [42,87,94,95,96,97,98,99,100,101,102,103,104,105,106,107,108,109,110]. One study used both 1.5-Tesla and 3-Tesla MRI systems (Table 1, SN 4) [74], and one study did not specify the MRI system used (Table 1, SN 15) [111].

Furthermore, PET techniques varied depending on the study’s objectives. Previous work employed PET scanners from different vendors such as General Electrics (GE) Advance PET/computed tomography (CT) (n = 6; Table 1, SN 3, 5, 6, 11, 12, 17) [42,97,99,100,103,104], GE Discovery PET/CT (n = 6; Table 1, SN 6, 11, 12, 15, 17) [42,99,100,102,103,111], and Siemens Biograph HiRez XVI (n = 2), Siemens Biograph mCT (n = 4; Table 1, SN 1, 2, 18, 20) [95,96,101,107], Siemens Biograph mMR (n = 1; Table 1, SN 19) [94], GE SIGNA PET/MRI (n = 2; Table 1, SN 10, 16) [87,98], and GE Healthcare PET/CT (n = 5; Table 1, SN 7, 8, 11, 13, 14) [105,106,108,109,110], while one system remained undefined in a study (Table 1, SN 4) [74]. Attenuation correction techniques, including low-dose CT scans and 68Ge transmission scans, were commonly integrated to enhance image precision. Radiotracers like [18F]AV-1451 for tau pathology, [11C]PK11195 for TSPO, and [11C]UCB-J for synaptic density measurements targeted specific biological markers. The PET data were prone to rigorous processing, encompassing motion correction, and correction for partial volume effects (PVE), culminating in SUVRs to quantitatively assess radiotracer concentration in distinct brain regions. Temporal data capture occurred via multiple frames, while spatial resolution was optimized with voxel sizes.

#### 3.2.1. Findings from PET/CT with MRI Co-Registration

First, [18F]AV-1451 (also known as [18F]-T807) was used in nine of the identified studies (Table 1, SN 1, 2, 3, 4, 7, 8, 9, 12, 14, 15) [42,74,95,96,104,106,108,109,110,111]. Furthermore, [11C]PK11195 was used in two previous studies (Table 1, SN 11, 17) [99,100], while [11C]UCB-J was mentioned in Holland et al. (Table 1, SN 10) [87] and Mak et al. (Table 1, SN 16) [98]. Moreover, 18F-FDG was used in Seckin et al. (Table 1, SN 13) [105], while [18F]PM-PBB3 was mentioned in Tagai et al. (Table 1, SN 18) [107] and Matsuoka et al. (Table 1, SN 20) [101]. Use of [18F]RO948 was mentioned in Oliveira Hauer et al. (Table 1, SN 21) [102], and [18F]PI-2620 was investigated in Aghakhanyan et al. (Table 1, SN 19) [94].

##### [18F]AV-1451

The application of [18F]AV-1451, a tracer renowned for its strong affinity to tau pathology, was used in nine studies (Table 1, SN 1, 2, 3, 4, 7, 9, 11, 14, 17) [74,95,96,99,100,104,108,109,110]. These investigations have unveiled compelling results, particularly in subcortical regions, with the striatum demonstrating robust signals across all groups. Notably, significant atrophy was detected in PSP, with the caudate, thalamus, and parietal lobe displaying the highest standardized SUVRs [96]. Of note, despite elevated SUVRs in PSP compared to HC, Coakeley et al. could not observe notable distinctions in SUVRs among PSP, PD, and HC [96]. This study underscored the potential of [18F]AV-1451 in detecting tau pathology while also underscoring the considerable challenge of distinguishing PSP from other diseases [96]. These findings were partially confirmed by another study by Coakeley et al., which documented a lower uptake of [18F]AV-1451 in the substantia nigra in both PSP and PD patients compared to HC [95]. This result suggested a promising utility in discerning these conditions from HC; however, the study did not identify any significant differences in mean uptake between PSP and PD [95].

In contrast, Passamonti et al. took a different approach by using [18F]AV-1451 [104]. Their results showed increased [18F]AV-1451 binding in AD and PSP compared to HC [104]. Notably, the binding patterns showed specificity to different brain regions, with AD showing increased binding in cortical and subcortical areas, whereas PSP showed increased binding in regions such as the midbrain, putamen, pallidum, thalamus, and dentate nucleus, which highlighted the potential of [18F]AV-1451 to effectively discriminate between these two conditions [104]. Also, other studies reported significantly elevated [18F]AV-1451 uptake in various regions of the brain in PSP patients when compared to both PD and HC [74,108,110]. Specifically, regions such as the globus pallidus, putamen, subthalamic nucleus, midbrain, precentral cortex, and dentate nucleus exhibited marked increases in tracer uptake among PSP patients [74,108,110]. These results may indicate the promise of [18F]AV-1451 in effectively distinguishing PSP from other neurodegenerative disorders [74].

In a related study by Whitwell and colleagues, the researchers explored the intricate landscape of PSP variants [109]. The results unveiled distinctive patterns of involvement among these variants: while common areas like the striatum, thalamus, anterior hippocampus, and putamen were implicated, it became evident that different PSP variants exhibited their unique signatures in terms of affected brain regions [109]. In the “classical” PSP variant Richardson’s Syndrome (PSP-RS), elevated flortaucipir uptake was observed in the pallidum, putamen, thalamus, subthalamic nucleus, and red nucleus, and, additionally, increased uptake was noted in the caudate, midbrain (specifically the superior aspects of the substantia nigra), dentate nucleus of the cerebellum, and cerebellar GM [109]. In the PSP-Speech/Language (PSP-SL) variant, there was increased flortaucipir uptake in the pallidum, putamen, thalamus, and the frontal lobe, with greater uptake specifically in the left hemisphere [109]. A milder uptake was observed in the subthalamic nucleus, midbrain (including the red nucleus and substantia nigra), and cerebellar GM [109].

For both the PSP-Predominant Parkinsonism (PSP-P) and PSP-Progressive Gait Freezing (PSP-PGF) variants, more focal patterns of uptake were observed, primarily in the putamen and pallidum, with some additional regions of elevated uptake identified at a less stringent statistical threshold [109]. Conversely, in the PSP-Corticobasal Syndrome (PSP-CBS) variant, no regions of elevated flortaucipir uptake were found compared to controls [109].

##### [11C]UCB-J

Holland et al. used [11C]UCB-J PET imaging to study synaptic density in PSP and CBD patients compared to HC, aiming to assess its diagnostic and disease progression monitoring potential for tauopathies [87]. Their findings showed significant widespread reductions in [11C]UCB-J (indicating synaptic density) in PSP and CBD patients compared to HC across various brain regions, both cortically and subcortically [87]. In PSP patients, the most pronounced synaptic density reductions were localized in the medulla, substantia nigra, pallidum, midbrain, pons, and caudate nucleus [87,98]. In CBD patients, significant reductions occurred in the medulla, hippocampus, amygdala, caudate nucleus, insula, and thalamus. Notably, these synaptic losses were observed even in regions without significant GM atrophy, demonstrating the sensitivity of [11C]UCB-J PET in the identification of early synaptic changes preceding structural alterations [87].

##### [11C]PK11195 and [18F]AV-1451

In the study by Passamonti et al., the results showed increased [11C]PK11195 binding in the thalamus, putamen, and pallidum when comparing PSP patients with HC [104]. Importantly, significant correlations were found between [11C]PK11195 binding and cognitive phenotype/performance and disease severity, highlighting its potential as a diagnostic and monitoring marker for PSP [103,104].

A study by Malpetti and colleagues further investigated this relationship between [11C]PK11195 binding and [18F]AV-1451 binding in PSP, showing a significant correlation across the brain [99]. These PET markers were observed to be associated with baseline disease severity and predicted subsequent clinical progression [100]. Specific brain regions with high [11C]PK11195 binding included the brainstem, cerebellum, thalamus, and various cortical regions in PSP patients, while high [18F]AV-1451 binding was observed in the basal ganglia, midbrain, and thalamus [100]. Furthermore, significant positive correlations were reported between corresponding components of [11C]PK11195 and [18F]AV-1451 binding in specific brain regions, particularly motor-related regions in PSP patients [100]. This may highlight the potential of [11C]PK11195 PET imaging, shedding light on both neuroinflammatory components and their relationship with tau pathology and clinical outcomes.

##### [18F]RO948

In a study on PSP using [18F]RO948 PET imaging, the authors found distinct patterns of tracer uptake in different brain regions, with PSP patients having higher SUVRs in the globus pallidus and lower SUVRs in the substantia nigra, distinguishing them from other neurodegenerative diseases such as dementia with Lewy bodies (DLB) and PD [102]. The combination of multiple imaging and neurochemical biomarkers, including PET imaging, MRI midbrain-to-pons area ratio (M:P), and CSF neurofilament light (NfL) levels, significantly improved the accuracy of diagnosis and differentiation of PSP from other neurodegenerative syndromes [102].

##### [18F]-PM-PBB3

A different methodological approach was taken by Tagai et al., who performed 18F-PM-PBB3 imaging on the PET/CT and then co-registered it with an MRI dataset [107]. The authors focused on improving the quantification of tau deposits by defining reference voxels in GM and WM regions using Gaussian distributions to distinguish pathological voxels [107]. Among the reference tissues studied (i.e., cerebellum, GM, and WM), GM stood out for its robust diagnostic performance across all diseases and age factors [107]. This result may be particularly important in diseases such as PSP and CBD. Since the reliability of GM assessment for quantifying tau deposits using 18F-PM-PBB3 PET imaging suggests that it might be favored as a reference tissue, it can help improve the accuracy of the tau pathology assessment and advance PET imaging techniques for tau-related diseases [107].

#### 3.2.2. Findings from PET/MRI

##### [18F]PI-2620

In Aghakhanyan’s study, 24 individuals with PSP underwent [18F]PI-2620 PET/MRI imaging to assess tau burden [94]. The study revealed that tau load in specific brain regions, such as the right globus pallidus externus and left dentate nucleus, was associated with alterations in region-to-region functional connectivity as assessed by resting-state functional MRI (fMRI) [94]. Notably, the network related to an increased tau load in the right globus pallidus externus exhibited hyperconnectivity in low-range intra-opercular connections, while the network linked to increased tau load in the left cerebellar dentate nucleus showed cerebellar hyperconnectivity and cortico-cerebellar hypoconnectivity [94,106]. These findings pointed to aberrant connectivity, as evidenced by a significant network-based statistic (NBS) network in PSP compared to HC, consisting of 89 regions of interest (ROIs) and 118 connections [94]. The study demonstrated the significant impact of tau load on functional network connectivity, thus shedding light on the synergistic effects of tau pathology on brain networks in PSP [94].

##### [11C]UCB-J

In the study by Holland et al., a comprehensive neuroimaging approach was employed to investigate PSP and CBD in comparison to HC [87]. Specifically, PET imaging using [11C]UCB-J revealed widespread reductions in synaptic density in various brain regions of PSP and CBD patients, even in regions without significant GM atrophy [87]. Notable reductions were seen in specific regions, such as the medulla, substantia nigra, pallidum, midbrain, pons, and caudate nucleus in PSP, and the medulla, hippocampus, amygdala, caudate nucleus, insula, and thalamus in CBD [87,98]. Correlations were established between [11C]UCB-J binding potential and cognitive performance, highlighting its potential role as a diagnostic and disease progression marker [87]. Meanwhile, MRI analysis demonstrated significant cortical thinning and subcortical atrophy in specific regions, extending beyond areas affected by atrophy. These findings emphasize the synergy of assessing both synaptic density and structural changes in tauopathies, offering a comprehensive understanding of disease progression [87,98].

In the study by Mak et al., the analysis integrated both PET/MRI findings to provide a comprehensive understanding of the synergistic effect in the context of PSP and CBD. The PET imaging component revealed significant global reductions in the binding potential of [11C]UCB-J, indicative of synaptic density, in patients with PSP and CBD compared to HC [98]. These reductions encompassed various cortical and subcortical brain regions, with specific areas affected, such as the medulla, substantia nigra, pallidum, midbrain, pons, and caudate nucleus in PSP, and the medulla, hippocampus, amygdala, caudate nucleus, insula, and thalamus in CBD [98]. Importantly, the study highlighted that synaptic loss was evident even in brain regions not exhibiting significant GM atrophy, underlining the potential of synaptic PET as a valuable approach for diagnosis and disease progression monitoring [98]. The MRI findings utilizing T1w imaging supplemented the PET results, given that PSP patients displayed significant cortical thinning and subcortical atrophy, particularly in the motor cortex, frontal cortices, thalamus, putamen, pallidum, and midbrain [98]. In CBD patients, focal cortical thinning was observed in the motor cortex, superior frontal cortex, and occipital lobe, alongside subcortical atrophy in the left putamen and bilateral pallidum [98]. Notably, cortical thinning extended beyond regions affected by atrophy, encompassing the temporo-parietal and cingulate cortices [98]. This integration of PET and MRI findings highlights the intricate relationship between structural and synaptic changes in PSP and CBD, offering insights into early pathological alterations in these tauopathies and emphasizing the value of a dual-modality approach for a more comprehensive understanding of disease progression [98].

#### 3.2.3. Findings from PET with Multi-Modal MRI

Only six of the included studies reported on the application of additional (advanced) MRI sequences in addition to anatomical imaging with T1w or PDw sequences (Table 1, SN 5, 8, 12, 16, 19, 20) [42,94,97,98,101,106]. Cope et al. used resting-state fMRI in a cohort of AD patients, PSP patients, and HC in conjunction with whole-brain graph-theoretical analysis to systematically investigate global and local network characteristics (such as nodal connectivity strength and other metrics) [97]. A major finding was that strongly connected nodes displayed more tau pathology in AD, independently of intrinsic connectivity, and nodes that accrued pathological tau were those that displayed graph metric properties associated with increased metabolic demand and a lack of trophic support rather than strong functional connectivity in PSP [97]. Hence, PSP appeared to be linked to a small number of subcortical structures, and increased tau burden in the midbrain and deep nuclei was associated with strengthened cortico–cortical functional connectivity in PSP [97]. 

Furthermore, a study by Aghakhanyan et al. also used resting-state fMRI in PSP patients and non-AD controls (with mild cognitive impairment), with the aim of extracting seed-based connectivity measures and calculating the NBS as a cluster-level technique based on the graph theoretical concept of connected components [94]. They found aberrant connectivity patterns as indicated by a significant NBS network in PSP compared to controls, with significant effects of tau load on functional network connectivity in the right globus pallidus externus and left dentate nucleus [94]. Associations of the network linked with increased tau load in the right globus pallidus externus and hyperconnectivity of low-range intra-opercular connections were revealed, as well as associations in the network linked with increased tau load in the left dentate nucleus and cerebellar hyperconnectivity and corticocerebellar hypoconnectivity [94]. Hence, tau burden seems to align with alterations in functional connectivity and network characteristics in PSP, thus providing a link between accumulation and distribution of tau and the brain’s synchronization and functioning [94,97].

Furthermore, DWI was employed by three of the studies (Table 1, SN 8, 12, 16) [42,98,106]. Sintini et al. used DWI for assessment of fractional anisotropy (FA) and mean diffusivity (MD) of WM tracts in the context of diffusion tensor imaging (DTI) modeling among PSP patients and HC [106]. The authors found significant associations in PSP between increased flortaucipir uptake and decreased FA and increased MD, respectively, in the superior cerebellar peduncle, sagittal stratum, and posterior corona radiata in PSP, and significant associations between decreased FA and increased MD in the body of the corpus callosum and anterior and superior corona radiata and volume loss in the frontal lobe [106]. In a similar approach, Nicastro and colleagues reported significantly decreased FA, increased MD, and radial diffusivity (RD) for PSP in the corpus callosum, bilateral internal capsule, corona radiata, posterior thalamic radiations, cingulate WM, superior longitudinal fasciculus, sagittal stratum, and uncinate fasciculus in PSP, while higher 18F-AV1451 binding significantly correlated with GM volume loss in frontal regions, as well as DTI changes in motor tracts, and cortical thinning in parieto-occipital areas, as obtained from DTI modeling in PSP patients and HC [42]. Moreover, there was an association between higher PSP-RS scores and increased RD in the WM close to the bilateral middle temporal gyrus, right lateral occipital gyrus, and cuneus, thus indicating a relationship between the brain’s microstructure and symptoms [42]. In the study by Mak et al., patients with either PSP or CBD and HC underwent DWI with an orientation dispersion index (ODI) modeling approach, which revealed that significant decreases in cortical ODI were more widespread than areas affected by atrophy in PSP compared to HC, and those widespread ODI decreases were identified even beyond the atrophy-affected regions [98]. Specifically, reductions in cortical ODI and [11 C]UCB-J non-displaceable binding potential were found in excess of atrophy in PSP and CBD as compared to HC [98]. Furthermore, regional cortical ODI was significantly associated with [11 C]UCB-J binding potential in disease-associated regions for PSP and CBD [98]. Therefore, spatially resolved microstructural changes related to PSP may overlap with binding profiles of tracers according to diffusion MRI [98].

A study by Matsuoka et al. applied single-voxel MRS with placements of voxels of interest (VOIs) in the anterior as well as posterior cingulate cortex among patients with PSP and HC [101]. This study found that glutathione levels of the posterior cingulate cortex were associated with apathy scales and tau depositions in the angular gyrus, although PSP cases did not show glutathione level alterations compared with HC [101]. Hence, synergistic contributions of tau pathologies and glutathione metabolism may exist in parallel with associations to apathy scales [101].

## 4. Discussion

The objective of this review was to provide an overview on studies using combined PET and MRI in PSP, including studies extracted from a systematic literature search according to the PRISMA guidelines and published in or after 2017. Overall, 21 studies were finally included. In the context of assessing tau pathology in neurodegenerative diseases, various PET tracers and MRI techniques have provided valuable findings.

The [18F]AV-1451 tracer, known for its strong affinity to tau pathology, has been a focus in multiple studies [42,95,108,110]. These investigations consistently revealed robust signal in subcortical brain regions, particularly in the striatum [42,110]. Notably, PSP patients exhibited significant atrophy, with the caudate, thalamus, and parietal lobe displaying the highest SUVRs [95,96]. However, distinguishing PSP from other conditions like PD remains challenging. Some studies suggested the utility in discerning PSP from HC, with lower uptake in the substantia nigra in both PSP and PD patients; however, difficulties arose in differentiating PSP from PD due to overlapping binding patterns given that there is comparable tracer uptake distribution in particular for brain regions linked with tau pathology, such as the midbrain and globus pallidus [43,72,74]. Specifically, PSP patients displayed increased tracer uptake in regions associated with PSP tau pathology, which partially overlapped with PD, particularly in subcortical areas [43]. The lack of specificity in ligand binding led to limitations in precisely quantifying 18F-flortaucipir (AV-1451) in subcortical structures, particularly impairing its usefulness for detecting PSP in its early stages [3,43]. On the contrary, other studies found elevated [18F]AV-1451 binding in various brain regions in PSP patients, making it a promising tool for distinguishing PSP from other neurodegenerative disorders [108,110]. Furthermore, when exploring different PSP variants, this tracer unveiled unique patterns of involvement, indicating that distinct variants may exhibit their signatures in terms of affected brain regions [42,110]. Overall, [18F]AV-1451 shows promise in detecting tau pathology, but distinguishing PSP from other conditions remains challenging.

Imaging with [11C]UCB-J PET allowed the study of synaptic density in PSP and CBD patients [87,98]. The findings consistently demonstrated significant reductions in synaptic density in both patient groups compared to HC, spanning various brain regions, both cortically and subcortically, and it seems striking that these reductions were observed even in regions without significant GM atrophy, most likely indicating the tracer’s sensitivity in identifying early synaptic changes that precede structural alterations [87,98]. This aspect highlights the potential of [11C]UCB-J PET as a valuable diagnostic and disease progression marker for tauopathies.

Studies employing [11C]PK11195 (i.e., TSPO imaging) revealed increased binding in the thalamus, putamen, and pallidum when comparing PSP patients with HC, and significant correlations were found between [11C]PK11195 binding and cognitive performance, as well as disease severity, highlighting its potential as a diagnostic and monitoring marker for PSP [99,103]. Additionally, a study found a significant correlation between [11C]PK11195 binding and [18F]AV-1451 binding, emphasizing a role for understanding the relationship between neuroinflammation and tau pathology in PSP [99,104].

In a study using [18F]RO948 PET imaging, distinctive patterns of tracer uptake were observed in different brain regions in PSP patients [102]. These patients exhibited higher SUVRs in the globus pallidus and lower SUVRs in the substantia nigra, distinguishing PSP from other neurodegenerative diseases like DLB and PD [102]. The combination of multiple imaging and neurochemical biomarkers, including PET imaging, MRI, M:P, and CSF NfL levels, significantly improved the accuracy of diagnosis and differentiation of PSP from other neurodegenerative syndromes [102]. Furthermore, a unique methodological approach used [18F]PM-PBB3 imaging to improve the quantification of tau deposits [107]. Reference voxels in GM and WM regions were defined, enhancing the accuracy of tau pathology assessment and advancing PET imaging techniques for tau-related diseases [107]. Notably, GM showed robust diagnostic performance across all diseases and age factors, making it a favored reference tissue [107].

A study employing [18F]PI-2620 PET/MRI unveiled the association between tau load in specific brain regions and alterations in functional networks [94]. Specifically, PSP patients displayed aberrant connectivity in the right globus pallidus externus and left dentate nucleus, indicating the significant impact of tau pathology on brain networks [94]. This highlights the synergistic effects of tau pathology on brain networks in PSP. Furthermore, other work integrated PET and MRI findings to provide a more comprehensive understanding of PSP and related tauopathies [94]. Such integrative approaches can reveal synaptic loss in brain regions without significant GM atrophy, emphasizing the importance of assessing both synaptic density and structural changes [94]. This study shed light on early pathological alterations in these diseases, offering insights into disease progression and highlighting the value of dual-modality approaches for a more comprehensive understanding of the disorders.

Intriguingly, only a few studies were identified that used both PET and MRI beyond the acquisition and analysis of anatomical imaging such as T1w and PDw [42,94,97,98,101,106]. In those studies, diffusion MRI was mostly applied alongside PET, which can assess regional microstructural alterations [112,113,114,115,116]. In this regard, DTI is by far the most widely applied modeling approach for diffusion MRI, which can enable the assessment of anisotropy and structural orientation that is relevant as water molecules diffuse very differently depending on tissue microarchitecture and integrity [112,113,116]. Commonly, quantitative measures representative of certain diffusion properties are extracted, such as FA (i.e., directional diffusion preference) or MD (i.e., molecular diffusion rate) [112,113,116]. Correspondingly, two of the three studies using DWI also applied the DTI model, revealing associations between tracer binding or uptake and alterations of such measures [42,106]. Importantly, an association between higher PSP-RS scores and increased RD has been shown for certain WM regions, thus providing a link between diffusion MRI findings and clinical symptoms [42]. The remaining study used ODI for the characterization of microstructure changes beyond the classical DTI model [98]. It is known that the DTI model has inherent limitations such as low spatial resolution relative to the underlying WM structure (with typical voxel sizes in the range of 2–3 mm), or a tensor model that assumes that all fibers within a voxel are well-described by a single orientation estimate [117,118,119]. Furthermore, the standard tensor models diffusion as a Gaussian distribution, which does not appear to reflect biological reality [117,118,119]. Hence, more advanced models and extracted parameters (e.g., ODI) may help to provide more accurate assessments of the WM architecture. Another perspective could be provided by free-water (FW) and FW-corrected FA maps from a bi-tensor model, given that the presence of FW (i.e., water molecules that are not restricted by the cellular environment and therefore do not display a directional dependence) can considerably bias diffusion indices and lead to reduced FA and increased MD values [120]. Indeed, elevated substantia nigra FW has been demonstrated for PD and AP [121]. Yet, beyond the substantia nigra, PSP, but not PD, showed a broad network of elevated FW and FW-corrected FA that included the basal ganglia, thalamus, and cerebellum [121]. An approach such as FW correction is particularly attractive, as it would work with standard diffusion MRI sequences that are nowadays widely acquired by the neuroimaging community (e.g., 64 gradient directions and b-values of 0/1000 s/mm^2^) and do not require too much scanning time when combined with imaging acceleration techniques.

The hybrid PET/MRI approach offers several strengths, including the simultaneous acquisition of PET and MRI data in a single procedure, providing a comprehensive dataset that has been derived from the same scanning session, hence restricting the time gap between information from different sequences to a minimum and delivering sequences that are in ultimately high spatial correspondence (if no repositioning and no movement of the patient has occurred) [122]. This concurrent acquisition allows for accurate attenuation correction related to detailed anatomical information from high-resolution MRI data. Additionally, improved differentiation of soft tissues contributes to more precise attenuation correction values. Specifically, concurrent acquisition with PET/MRI facilitates a more precise attenuation correction than techniques that solely rely on PET or less comprehensive anatomical data [122]. However, challenges include potential artifacts arising from interactions between PET and MRI methods, such as magnetic susceptibility artifacts, attenuation correction mismatches, and distortions in PET images due to the magnetic field during simultaneous acquisition [123,124,125]. These factors may introduce inaccuracies and compromise the overall reliability of the imaging data. The integration of these technologies also involves complex instrumentation and is associated with higher costs compared to stand-alone systems [126].

In contrast, the separate PET and MRI approach with subsequent co-registration also contains several advantages and disadvantages. Independent attenuation correction is a notable advantage, potentially leading to higher accuracy. Reduced artifacts result from the lower likelihood of interactions between PET and MRI technologies, providing flexibility in the study design with different imaging protocols for each modality. However, challenges arise from temporal differences between separate PET and MRI scans, leading to potential inaccuracies in attenuation correction and other postprocessing steps [126]. Motion correction—a crucial process for aligning dynamic PET data with static MRI images—becomes challenging due to temporal disparities. Furthermore, variations in physiological parameters such as changes in blood flow or tissue density between scans, can impact the accuracy of correction methods [124]. These temporal discrepancies may introduce uncertainties in the alignment of anatomical structures, potentially compromising the effectiveness of corrections and leading to errors in the final imaging results. Achieving accurate alignment of anatomical structures between PET and MRI data can be challenging, introducing the possibility of errors [127]. In cases where CT is used for attenuation correction in PET, there may be increased radiation exposure compared to the hybrid PET/MRI approach. Additionally, stand-alone PET may have limitations in differentiating soft tissues compared to the hybrid approach [126]. While both approaches have their strengths and limitations, the choice between hybrid PET/MRI and separate PET and MRI with co-registration depends on factors such as specific clinical requirements, available technologies, and the nature of the imaging study.

Moving on to the specific correction techniques in the domain of attenuation correction, the hybrid PET/MRI approach excels by allowing simultaneous acquisitions of PET and MRI data, leveraging enhanced accuracy from detailed anatomical information in MRI data. The notable absence of a radiation source for attenuation correction is a benefit when compared to PET/CT, which becomes especially relevant in repetitive examinations or for examinations in patients at younger age [122,126,127]. However, challenges include potential artifacts from PET and MRI technology interactions. Conversely, separate PET and MRI acquisitions with co-registration provide independent and potentially more robust attenuation correction for each modality system, hence likely improving accuracy. Nevertheless, challenges such as temporal differences between PET and MRI scans and the need for an additional radiation source (from CT) may persist. Attenuation correction remains a major methodological challenge in integrated PET/MRI systems, necessitating MRI-based approaches that account for bone tissue, lung, and MRI hardware impact on PET fields [125]. Issues related to subject motion, truncation, and susceptibility artifacts are addressed, with standard MRI-based techniques discussed alongside advanced methods, including deep learning (DL) approaches [125]. Specifically, DL approaches have demonstrated potential in mitigating attenuation correction challenges by leveraging their ability to learn complex relationships between MR images and corresponding attenuation maps. These methods offer promise in improving the accuracy of attenuation correction, especially in scenarios where traditional techniques may fall short [128]. In terms of partial volume correction, the hybrid PET/MRI strategy offers potential advantages with its simultaneous data acquisition. It enables a more accurate estimation of partial volumes by using high-resolution MRI data for precise anatomical definitions, which becomes especially relevant for tissue segmentations and ROI definitions (especially when it comes to rather small brain structures) [129]. In case of separate PET and MRI with joint registration of datasets, independent partial volume correction for each modality system is a potential strength that can lead to higher accuracy. However, challenges include difficulties in accurately aligning anatomical structures between PET and MRI data, which can lead to partial volume correction errors similar to attenuation correction [130]. Temporal differences between the two modalities further complicate the correction process. In motion correction, the hybrid PET/MRI approach with simultaneous acquisition offers strengths in reducing motion artifacts and utilizing MRI data for precise correction. However, challenges include artifacts from technology interactions and MRI sensitivity to movements. Subject motion, particularly from breathing and cardiac activity, is a recurring challenge in imaging studies, and movement of the patient may become more likely with more extensive imaging protocols [124]. While periodic motion can be controlled using external devices, unpredictable motion poses difficulties. External devices have limitations in characterizing complex internal organ motion, but PET and MRI data can provide detailed information for motion modeling. Integrating these methods into a workflow for routine clinical studies is challenging, but holds potential for improving image quality and reducing motion-related artifacts. In addition, with modern MRI software, parallel imaging, spiral imaging, or even image reconstruction using artificial intelligence can be realized, which can considerably decrease necessary acquisition times while providing high-quality images in general [131,132,133,134,135]. Hence, multi-sequence imaging protocols can become more and more feasible, given that, with such imaging techniques, more pulse sequences could be principally acquired in a limited amount of time.

While both radiological and radiopharmaceutical biomarkers have shown promise in PSP diagnosis, they each have limitations and may not provide sufficient diagnostic accuracy on their own [15]. Combining multiple biomarkers may offer a more comprehensive approach to PSP diagnosis [102,106]. The advantage of combining radiological and radiopharmaceutical biomarkers for PSP diagnosis is that the combination provides complementary information about the disease process and structural and functional brain changes associated with the disease [18]. For example, the next-generation tau PET tracer [18F]PI-2620 and resting-state fMRI have been used to assess functional connectivity and network alterations in PSP patients [94,136]. In these studies, PSP patients showed altered functional connectivity and tau load in specific brain regions in relation with changes in functional networks [94,136]. Such an approach can provide valuable information about the downstream effects of neuropathology on brain functionality in PSP [136]. In the analysis of the combination of PET and MRI, standardized tools could be applied like the MR Parkinsonism Index (MRPI) [137]. This volumetric method has a high sensitivity and specificity and thus facilitates individual differentiation, which offers a benefit in distinguishing patients with PSP from those with PD and MSA-Parkinsonism at an individual level [137,138]. After its first description in 2008 [137], automated MRPI calculation techniques have been successfully applied to large multi-national datasets [139]. An automated new version called MRPI2.0 has been developed for further improvement of the standardized volumetry [140]. These approaches could be important to standardize the MRI measures in patients with PSP across centers, with a positive impact on multicenter studies and clinical trials.

Despite progress in the development of biomarkers for PSP diagnosis, several challenges and limitations remain. One major challenge is the complexity of PSP pathophysiology, which involves the accumulation of tau protein and degeneration of multiple brain regions [15,141]. The intricate nature of PSP and related neurodegenerative disorders has indeed posed challenges in pinpointing specific biomarkers that can reliably distinguish PSP from other conditions. The complex pathological overlaps and clinical similarities among these disorders have contributed to this difficulty [142,143]. Additionally, there is considerable heterogeneity in the clinical presentation of PSP, which can further complicate diagnosis when considered in combination with imaging. There are several technical issues associated particularly with PET imaging that can impact the quality and reliability of results [130,144]. For example, image resolution can be limited, particularly for small structures such as the hippocampus [145]. Any PVE can also be a challenge where the signal from a small ROI is contaminated by activity from surrounding areas [74,99,100,146,147], implying the risk for underestimation of the signal and false-negative results [148]. Several approaches, including the geometric transfer matrix, enhanced iterative reconstructions, improved MRI-based attenuation correction, and point spread function modeling, can significantly enhance PET image quality, thereby reducing and addressing PVE [149,150,151,152]. Additionally, head motion during scanning can also impact the accuracy of PET images, as well as the quality of MRI data when multi-modal approaches with longer scan times are planned. Further, combined biomarker approaches for PSP diagnosis can be hampered by the costs and availability of technology used for biomarker testing, as especially combined PET/MRI systems may not be widely accessible in clinical settings. Despite advancements in imaging pulse sequence protocols as well as tracer developments, it is crucial to emphasize that, at the current stage, none of the available imaging modalities may be able to differentiate PSP from PD in individual patients with very high specificity [17,153,154]. While group data may reveal statistically significant differences in terms of differentiation, caution is warranted in clinical practice, where individual diagnosis remains a complex challenge and where patients present with a broad spectrum of disease stages [5]. Relying solely on group data for clinical decision-making may not accurately reflect the intricacies of individual cases. Recognizing the limitations of exclusive reliance on group data, an imperative shift toward more nuanced diagnostic strategies becomes evident. Particularly, integrating multi-modal approaches and combining various methods and sequences emerges as a promising avenue to enhance the precision and effectiveness of differentiation at the individual level [155].

An additional limitation arises from the in vivo reference standard of clinical criteria for syndrome diagnosis, which, while effective, is not as accurate as post-mortem histopathological examinations. This introduces the possibility of misdiagnosed cases, where individuals with PSP may be incorrectly diagnosed as having PD, thus entering imaging studies with the correct diagnosis in question [156]. Acknowledging this limitation is crucial for interpreting and contextualizing imaging results accurately. Correlation of in vivo imaging findings to findings from histopathology remains essential, serving as the ultimate reference standard for refining and validating imaging findings, especially in the dynamic landscape of neurodegenerative diseases [157]. Future research into PSP biomarkers should prioritize the development of multimodal biomarker methods that integrate clinical and imaging biomarkers. These approaches should be validated in large, multi-center cohorts of patients with PSP and other neurodegenerative disorders to ensure their accuracy and reliability. Further research is also needed to identify novel biomarkers that can provide insight into the underlying pathophysiology of PSP, including genetic and molecular biomarkers. Finally, it is important to develop biomarkers that can accurately predict disease progression and treatment response in PSP, which can aid in the development of personalized treatment strategies for patients with PSP.

## 5. Conclusions

Radiopharmaceutical biomarkers and PET imaging have emerged as a valuable tool in the study of neurodegenerative diseases like PSP, particularly in the imaging of tau protein aggregates. However, there are several challenges associated with tau PET imaging, including off-target binding, inconsistencies between ante- and post-mortem findings, and technical limitations. When combined with multiparametric and especially quantitative MRI, a more comprehensive and accurate approach to PSP diagnosis could be realized that is based on the complementary use of metabolic, macro- and microstructural, and functional information. However, only a few studies have been published over recent years that employed MRI sequences beyond anatomical T1w imaging.

In the future course for PSP diagnosis and its integration into clinical practice, it is imperative to explore biomarkers that offer specificity and precision. These innovative biomarkers encompass various categories, including diagnostic biomarkers, facilitating early disease identification; monitoring biomarkers, adept at tracking disease progression; response biomarkers, enabling the assessment of intervention efficacy; predictive biomarkers, offering insights into patient outcomes; and prognostic biomarkers, shedding light on the disease course. These biomarkers should not only advance our comprehension of the underlying disease pathology but also play a pivotal role in expediting early clinical interventions and for the development of more efficacious treatment strategies [133]. Future directions for developments in PSP diagnosis should involve the identification of new biomarkers that can provide additional information about the underlying pathology of the disease and could potentially facilitate earlier intervention and treatment.

## Figures and Tables

**Figure 1 cells-12-02776-f001:**
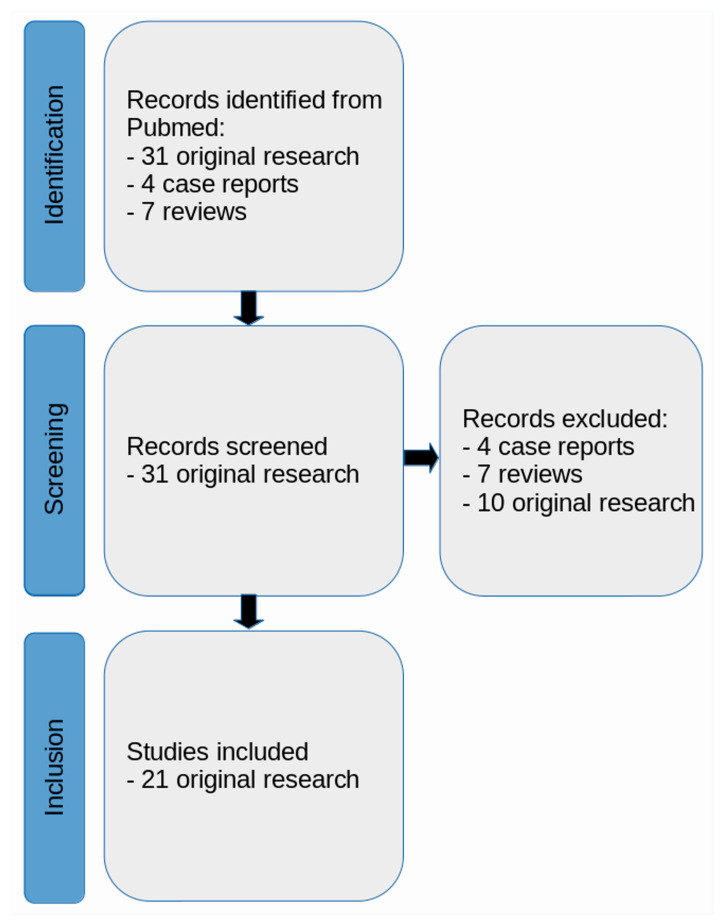
PRISMA flow diagram illustrating the literature review and study selection process. In total, 21 original research articles were included in the present review.

**Figure 2 cells-12-02776-f002:**
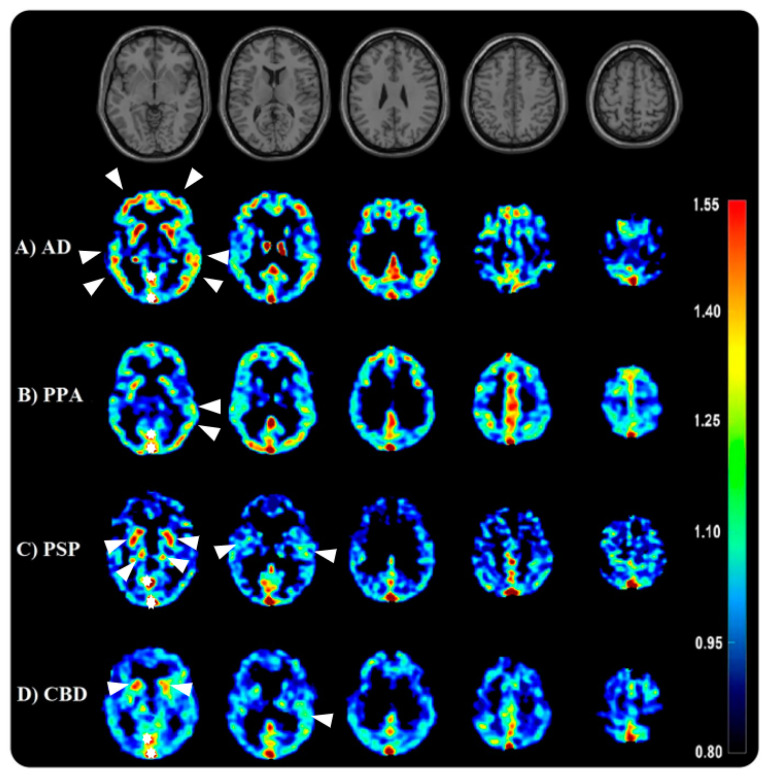
Serial transverse brain sections from [11C]PBB3-PET for (**A**) a patient with Alzheimer’s disease (AD: 73 years old), with tau distribution in medial and lateral temporal cortices as well as the frontal cortex, (**B**) a patient with logopenic variant primary progressive aphasia (IvPPA: 69 years old), with tau distribution in left temporal medial areas, (**C**) a patient with progressive supranuclear palsy (PSP: 56 years old), with tau distribution in frontoparietal WM, parietal GM, globus pallidus, and subthalamic nucleus, and (**D**) a patient with corticobasal degeneration (CBD: 52 years old) with tau distribution in basal ganglia and frontotemporal areas. The single-subject T1-weighted structural template is also presented to provide a detailed view of the brain structures. Note the off-binding signals in (**A**–**D**), such as the dural sinus indicated by the star. The white arrows show the enhanced tau uptake.

**Table 1 cells-12-02776-t001:** 21 articles published between 2017 and 2023 after application of the inclusion and exclusion criteria. a—years (age given as mean ± standard deviation, if not otherwise stated); AUC—Area Under the Curve; AD—Alzheimer’s Disease; BPND—Non-Displaceable Binding Potential; CBD—Corticobasal Degeneration; CBS—Corticobasal Syndrome; CSF—Cerebrospinal Fluid; CT—Computed Tomography; DLB—Dementia with Lewy Bodies; DTI—Diffusion Tensor Imaging; FA—Fractional Anisotropy; f—Female; GE—General Electric; GM—Gray Matter; HC—Healthy Control; MCI—Mild Cognitive Impairment; MD—Mean Diffusivity; min—Minutes; MMSE—Mini-Mental State Examination; MoCA—Montreal Cognitive Assessment; MRI—Magnetic Resonance Imaging; MPRAGE—Magnetization Prepared Rapid Gradient Echo; NBS—Network-Based Statistics; NfL—Neurofilament Light; ODI—Orientation Dispersion Imaging; PD—Parkinson’s Disease; PET—Positron Emission Tomography; PSP—Progressive Supranuclear Palsy; PSPRS—Progressive Supranuclear Palsy Rating Scale; RAVLT—Rey Auditory Verbal Learning Test; ROI—Region of Interest; SUV—Standardized Uptake Value; SUVR—Standardized Uptake Value Ratio; UPDRS—Unified Parkinson’s Disease Rating Scale; WM—White Matter.

Author, Year and Study Number (SN)	Study Cohort	Radio- Pharmaceutical	Image Acquisition	Main Findings
Coakeley 2017 [96] (1)	- 6 PSP (72.2 ± 6.8 a, 4 f) - 6 PD (63.7 ± 9.6 a, 3 f) - 10 HC (65.9 ± 9.9 a, 8 f)	[18F]AV-1451 (also known as [18F]-T807)	- High-resolution Siemens Biograph HiRez XVI (Siemens Healthcare, Erlangen, Germany) Low-dose CT scan for attenuation correction. 90 min PET scan with SUVs calculated between 60 to 90 min post-injection. Emission data were processed into 28 frames. SUVR were relative to the cerebellar cortex.	- In PET analysis, subcortical regions, particularly the striatum, showed signals across all groups. In PSP, atrophy was observed by MRI, with the caudate, thalamus, and parietal lobe exhibiting the highest SUVRs. In PD all regions had SUVRs greater than 1, indicating [18F]AV-1451 uptake above the cerebellar reference. No significant differences in SUVR in PSP compared to PD or HC. No correlation between PSPRS and UPDRS scores and [18F]AV-1451 uptake. MoCA scores were not predictors of SUVR in any of the ROIs analyzed. [18F]AV-1451 may bind selectively to paired helical filaments in AD, which differ from the straight conformation of tau filaments in PSP.
Coakeley 2018 [95] (2)	- 6 PSP (72.2 ±6.8 a, 4 f) - 6 PD (63.7 ± 9.6 a, 3 f) - 10 HC (65.9 ±9.9 a, 8 f)	[18F]AV-1451 ([18F]-T807)	- High-resolution Siemens Biograph HiRez XVI (Siemens Healthcare, Erlangen, Germany) Emission list mode data rebinned into 3D sinograms. PET acquisition from 0 to 90 min was binned into 28 frames.	- [18F]AV-1451 binds neuromelanin in the substantia nigra. Decrease in [18F]AV-1451 uptake in the substantia nigra of patient groups vs. HC. Significantly higher SUVR in HC compared to PSP and PD. No significant difference between mean uptake in PD compared to PSP.
Passamonti 2017 [104] (3)	- 15 AD (including amyloid-positive MCI) (71.6 ± 8.7 a, 6 f) - 19 PSP (69.5 ± 5.8 a, 8 f) - 13 HC (67.2 ± 7.3 a, 7 f) - Post-mortem brain tissue (1 AD, 1 PSP, 1 HC)	18F-AV-1451 & 11C-PiB radiotracer	- PET with GE Advance PET scanners. (GE Healthcare, Waukesha, WI, USA) MRI, CSF correction, and quantification of binding potential.	- 18F-AV-1451 BP_ND_ values distinguish with a classification accuracy of 94.1% between AD/MCI+ patients and PSP cases. The accuracy for other comparisons was 85.7% for AD/MCI+ vs. HC and 90.6% for PSP vs. HC. Significantly increased 18F-AV-1451 binding for AD and PSP vs. HC. Binding was significantly increased in AD (vs. PSP/HC) in the hippocampus, occipital, parietal, temporal, and frontal cortices. Binding was significantly increased in PSP (vs. AD) in the midbrain. Uptake was significantly increased in PSP (vs. HC) in the putamen, pallidum, thalamus, midbrain, and dentate nucleus. 18F-AV-1451 strongly bound to AD-related tau pathology, but less specifically in PSP in post-mortem samples.
Schonhaut 2017 [74] (4)	- 33 PSP (69.6 ± 5.7 a, 10 f) - 26 PD (67.1 ± 5.4 a, 12 f) - 46 HC (69.6 ± 5.4 a, 21 f) - Post-mortem brain tissue (1 clininal PSP/CBD)	18F-flortaucipir (formerly 18F-AV1451 or 18F-T807)	- PET scanner not specified. Multicenter study. SUVRs were calculated from PET data collected 80–100 min post-injection using a cerebellum gray matter reference region.	- Increased 18F-flortaucipir uptake for PSP in globus pallidus, putamen, subthalamic nucleus, midbrain, and dentate nucleus vs. PD and HC. Globus pallidus binding best distinguished PSP from PD and HC: AUC = 0.89 & 0.87. No correlation between PSPRS or MMSE scores and 18F-flortaucipir retention in any region. Severe tau pathology in PSP-related brain structures with correspondence between in vivo 18F-flortaucipir and post-mortem samples.
Cope 2018 [97] (5)	- 17 AD (71 ± 9 a) - 17 PSP (69 ± 6 a) - 12 HC (67 ± 8 a)	18F-AV-1451 and 11C-PiB	- GE Advance PET scanner (GE Healthcare, Waukesha, WI, USA) with a 15 min 68Ge transmission scan for attenuation correction. 11C-PiB scans were acquired with a bolus injection of 9.0 to 11.0 mCi, 60 min dynamic acquisition in 69 frames. Non-displaceable binding potential was assessed in specific ROIs (superior cerebellum as reference). MRI: resting-state fMRI.	- In AD, a correlation was observed between 18F-AV-1451 BPND and network connectivity, indicating that more strongly connected brain regions had higher tau accumulation (not present in PSP and HC). In AD, a strong positive correlation between 18F-AV-1451 BPND and node degree was observed. Conversely, in PSP, no strong positive correlations were observed in any network. In PSP, tau accumulation was more focal, with higher 18F-AV-1451 BPND in midbrain and basal ganglia. Increasing tau burden in the midbrain and deep nuclei was associated with strengthened cortico-cortical functional connectivity in PSP. Disrupted cortico-subcortical and cortico-brainstem interactions meant that information transfer took less direct paths.
Passamonti 2018 [103] (6)	- 16 AD (including amyloid-positive MCI) (68.7 ± 8.6 a, 7 f) - 16 PSP (68.4 ± 5.7 a, 6 f) - 13 HC (68.0 ± 5.3 a, 8 f)	[11C]PK11195 and 11C-PiB	- GE Advance PET scanner/GE Discovery 690PET/CT (GE Healthcare, Waukesha, WI, USA). Attenuation correction by a 15 min 68Ge/68Ga transmission scan and a low-dose CT scan. Injection of 550 MBq [11C]PiB and acquisition 40–70 min post-injection, 75 min dynamic imaging session (55 frames) starting simultaneously with a 500 MBq [11C]PK11195 injection.	- The PSP group vs. HC showed increased [11C]PK11195 BPND in the thalamus, putamen, and pallidum. In the AD/MCI+ group, there was a significant negative correlation between the RAVLT scores (delayed recall at 30 min) and [11C]PK11195 BPND in the precuneus. Significantly increased [11C]PK11195 in the medial temporal lobe and occipital, temporal, and parietal cortices in AD compared to PSP and HC. Significantly increased [11C]PK11195 binding in the thalamus, putamen, and pallidum in PSP compared to HC. Significant correlation between RAVLT scores and [11C]PK11195 binding in the cuneus/precuneus in AD. Significant correlation between PSPRS scores and [11C]PK11195 binding in the pallidum, midbrain, and pons in PSP.
Whitwell 2018 [108] (7)	- 30 PSP (median 70 a, range 65–73 a, 18 f) - 64 age- and gender-matched Aβ-negative HC	[18F]AV-1451 PET	- GE Healthcare PET/CT (GE Healthcare, Waukesha, WI, USA). A bolus injection of 370 MBq of [18F]AV-1451, followed by a 20 min PET acquisition 80 min after injection. Subjects had MPRAGE sequences performed within 48 h of the PET scans.	- Higher AV-1451 SUVR in AD-related regions (middle temporal gyrus, inferior temporal gyrus, fusiform, amygdala, entorhinal cortex and parahippocampal gyrus) was associated with higher AV-1451 SUVR in PSP-related regions (midbrain, globus pallidum, thalamus, dentate nuclei of the cerebellum and motor cortex). Higher PiB SUVR was not significantly associated with AV-1451 SUVR in AD- or PSP-related regions. PiB(−) group showed WM atrophy in midbrain and superior cerebellar peduncles, and GM atrophy in frontal regions and striatum vs. HC. PiB(+) group showed WM atrophy in midbrain and GM atrophy in frontal regions and striatum vs. HC. Aβ deposition occurs in PSP predominantly as an age-related phenomenon, with little evidence that Aβ and PHF-tau play a significant role in neurodegeneration in PSP.
Sintini 2019 [106] (8)	- 34 PSP (median 71 a, range 65–74 a, 19 f) - 29 HC (median 73 a, range 59–75 a, 17 f)	Flortaucipir PET scans-[18F]AV-1451 PET	- GE Healthcare PET/CT (GE Healthcare, Waukesha, WI, USA). A bolus injection of 333–407 MBq of flortaucipir, followed by a 20 min PET scan 80 min after injection. Four 5 min dynamic frames after a low-dose CT. MRI: DTI.	- Correlation between increased flortaucipir uptake in the cerebellar dentate, red nucleus, and subthalamic nucleus and decreased volume in the same regions in PSP. Correlation between increased flortaucipir uptake and decreased FA and increased MD in the superior cerebellar peduncle, sagittal striatum, and posterior corona radiata in PSP. Correlation between decreased FA and increased MD in the body of the corpus callosum and anterior and superior corona radiata and volume loss in the frontal lobe in PSP. The red nucleus was the region with the highest canonical weight for flortaucipir and the superior cerebellar peduncle had the highest canonical weight for DTI-MD.
Whitwell 2019 [110] (9)	- 16 PSP (68 ± 6 a, 6 f) - 39 age-matched HC (18 f)	[18F]AV-1451 tau-PET	- GE Healthcare PET/CT (GE Healthcare, Waukesha, WI, USA). Two serial [18F]AV-1451 PET scans with 12-month follow-up intervals on a GE Healthcare PET/CT. Bolus injection of 333–407 MBq and a 20 min PET scan 80 min after injection.	- Rate of midbrain atrophy was significantly increased in PSP vs. HC. [18F]AV-1451 regional change measures were significantly increased in PSP vs. HC in the pallidum, precentral cortex, dentate nucleus, and midbrain. Correlation of PSP Rating Scale with change in midbrain volume, but not with changes in the [18F]AV-1451 measures. No significant associations between baseline [18F]AV-1451 uptake and change in [18F]AV-1451 uptake over time.
Holland 2020 [87] (10)	- 14 PSP (72.8 ± 7.7 a, 7 f) - 15 CBD (70.6 ± 8.2, 2 f) - 15 HC (68 ± 7 a, 8 f)	[11C]UCB-J and 11C-PiB	- GE SIGNA PET/MR (GE Healthcare, Waukesha, WI, USA) Simultaneous 3T MRI and [11C]UCB-J (binds to synaptic vesicle glycoprotein 2A, a marker of synaptic density) PET imaging. Patients with CBS also underwent [11C]PiB PET.	- Synaptic density was quantified using [11C]UCB-J BPND. [11C]UCB-J binding was significantly decreased in frontal, temporal, parietal, and occipital lobes, cingulate, hippocampus, insula, amygdala, and subcortical structures in PSP and CBD compared to HC. Correlation between global [11C]UCB-J binding and PSP Rating Scale and CBD Rating Scale in PSP and CBD. Correlation with the revised Addenbrookes Cognitive Examination in PSP and CBD.
Malpetti 2020 [100] (11)	- 17 PSP (68.3 ± 5.7 a, 7 f) - 31 HC (68.6 ± 7.1 a, 15 f)	[11C]PK11195 and [18F]AV-1451 BPND	- GE Advance and a GE Discovery 690 PET/CT (GE Healthcare, Waukesha, WI, USA).	- Correlation between regional group mean [11C]PK11195 BPND and [18F]AV-1451 BPND across the whole brain in PSP. [11C]PK11195 binding was significantly higher in the putamen and pallidum in PSP compared to HC. [18F]AV-1451 binding was significantly higher in the putamen, pallidum, thalamus, midbrain, and dentate nucleus in PSP compared to HC. Principal Component Analysis identified four components for each ligand, reflecting the relative expression of tau pathology or neuro-inflammation in distinct groups of subcortical and cortical brain regions. Correlations between PSP-RS and subcortical tau pathology as well as neuroinflammation.
Nicastro 2020 [42] (12)	- 23 PSP (68.8 ± 5.8 a, 9 f) - 23 HC (68.7 ± 7.3 a, 11 f)	18F AV1451	- GE Advance or a GE Discovery 690 PET/CT (GE Healthcare, Waukesha, WI, USA), with a 15 min dynamic imaging following a 370 MBq 18F-AV1451 injection. 18F-AV1451 BP_ND_ was determined for specific ROIs. MRI: DTI.	- Significantly reduced GM volume in fronto-temporal regions, basal ganglia, midbrain, and cerebellum in PSP. Significantly reduced cortical thickness in left entorhinal and fusiform gyrus in PSP. Significantly decreased FA, increased MD and RD for PSP in the corpus callosum, bilateral internal capsule, corona radiata, posterior thalamic radiations, cingulate WM, superior longitudinal fasciculus, sagittal stratum, and uncinate fasciculus in PSP. Higher 18F-AV1451 binding significantly correlated with GM volume loss in frontal regions, DTI changes in motor tracts, and cortical thinning in parieto-occipital areas. Correlation between higher PSP-RS score and GM volume loss in bilateral middle cingulate gyrus, right middle temporal gyrus, left fusiform and middle occipital gyri, bilateral cuneus, and left cerebellum. Correlation between higher PSP-RS score and increased RD in WM facing bilateral middle temporal gyrus, right lateral occipital gyrus, and cuneus.
Seckin 2020 [105] (13)	- 8 PPAOS, developing Parkinsonian features (median 66 a, range 54–74 a, 3 f) - PSP and HC from another study	[18F]fluorodeoxyglucose	- GE Healthcare scanner (GE Healthcare, Waukesha, WI, USA) at baseline and the last visit. Injection of 18F-FDG after a 30 min uptake period, an 8 min 18F-FDG scan was performed.	- Progression was observed on FDG-PET in all eight patients, with hypometabolism especially in the premotor cortex, Decline in striatal hypometabolism was observed and was relatively symmetric. In four patients, greater change occurred in the left striatum compared to the right over time. Absent to minimal midbrain hypometabolism was observed across patients. Baseline midbrain volumes were similar to HC in all but one patient. Progression was observed by the time of the last visit, with three patients having volumes similar to PSP.
Whitwell 2020 [109] (14)	- 105 PSP (median 71 a, range 65–80 a, 53 f) - 30 HC - Post-mortem brain tissue (21 PSP)	[18F]flortaucipir ([18F]AV-1451)	- GE PET/CT GE Discovery LS PET/CT bolus injection of 370 MBq.	- Differences Across PSP Variants: Differences in neuroimaging findings were observed across different PSP variants. Consistent and Complementary Information: Both MRI and flortaucipir PET provided consistent and complementary information regarding the involvement of different brain regions in various PSP variants. Subcortical Circuitry Involvement: The study explored the subcortical circuitry typically affected in PSP and found that each PSP variant affects a different set of structures within this system. PSP-RS Findings: PSP-RS showed involvement of various subcortical structures, including the midbrain, striatum, subthalamic nucleus, and more. Disruptions in the dopaminergic nigrostriatal pathway were also noted. PSP-CBS and PSP-F Findings: PSP-CBS and PSP-F variants exhibited evidence of atrophy in the superior cerebellar peduncle, midbrain, striatum, globus pallidus, subthalamic nucleus, and thalamus. These variants also showed frontal cortical involvement. PSP-SL Findings: PSP-SL was characterized by atrophy and flortaucipir uptake in premotor and precentral (motor) cortex, which is typical for patients with progressive apraxia of speech. However, it showed less involvement of infratentorial structures. PSP-P and PSP-PGF Findings: PSP-P and PSP-PGF had more restricted involvement of subcortical circuits compared to PSP-RS. They exhibited volume loss and elevated flortaucipir uptake in the striatum, globus pallidus, and thalamus. Shared Pathophysiological Mechanisms: Some subcortical structures, such as the striatum, thalamus, and globus pallidus, were involved across all PSP variants, suggesting shared pathophysiological mechanisms.
Zhao 2020 [111] (15)	- 19 PSP (67.3 ± 9.1 a, 6 f) - HC	18F-FDG-PET/CT	- GE Discovery LS PET/CT GE Discovery LS PET/CT.	- Enlargement of 3rd ventricle in 15 cases, midbrain atrophy in 13 cases, cerebral cortex atrophy in 8 cases, and cerebellum atrophy in 3 cases, and “Hummingbird” sign visible in 11 cases. FDG uptake was decreased in the bilateral frontal cortex, striatum, and midbrain. 12 patients with PSP showed hypometabolism in the midbrain, and 9 patients in bilateral frontal lobes when compared to HC.
Mak 2021 [98] (16)	- 22 PSP (70.9 ± 8.7 a, 12 f) - 14 CBD (70.0 ± 7.9 a, 5 f) - 27 HC (69.0 ± 7.3 a, 11 f)	[11C]UCB-J PET.	- GE SIGNA PET/MR GE Discovery LS PET/CT for 90 min starting immediately. After [11 C]UCB-J injection (median injected activity: 351 ± 107 MBq). MRI: DWI/ODI.	- Significant cortical thinning particularly in the motor cortex and frontal cortices, and subcortical atrophy was found in the thalamus, putamen, pallidum, and midbrain in PSP as compared to HC. Significant reductions in cortical ODI were more widespread than areas affected by atrophy in PSP as compared to HC. Focal cortical thinning in motor cortex, superior frontal cortex, and the occipital lobe, as well as atrophy in the left putamen and bilateral pallidum in CBD as compared to HC. Widespread ODI reductions were found extending beyond the atrophy-affected motor cortices to other regions that were relatively preserved from atrophy, notably the temporo-parietal and cingulate cortices in CBD as compared to HC. Significant bilateral ODI reductions in the caudate and putamen where no significant GM atrophy was shown in CBD as compared to HC. Reductions in cortical ODI and [11 C]UCB-J non-displaceable binding potential in excess of atrophy in PSP and CBD as compared to HC. Regional cortical ODI was significantly associated with [11 C]UCB-J binding potential in disease-associated regions in PSP and CBD.
Malpetti 2021 [99] (17)	- 17 PSP (68.3 ± 5.7 a, 7 f)	[11C]PK11195 and [18F]AV-1451 (18F-flortaucipir)	- GE Advance and GE Discovery 690 PET/CT (GE Healthcare, Waukesha, WI, USA)	- Correlation between Principal Component Analysis components from [11C]PK11195 binding potential and [18F]AV-1451 binding potential in the brainstem and cerebellum and subsequent annual rate of change in the PSPRS. PCA-derived 11C]PK11195 binding potential and [18F]AV-1451 binding potential correlated with regional brain volume in the same regions. Subcortical GM volumes were not significantly correlated with subsequent clinical progression. Subcortical GM volumes were not significantly related to clinical severity at baseline.
Tagai 2022 [107] (18)	- 40 PSP (70.4 ± 6.4 a, 18 f) - 23 AD (66.2 ± 10.0 a, 12 f) - 40 HC (68.6 ± 5.7 a, 15 f) - 5 tau-positive FTLD	18F-PM-PBB3	- Biograph mCT Flow system (Siemens Healthcare, Erlangen, Germany), with 18F-PM-PBB3 radiotracer-generation of SUVR images with the cerebellum, GM and WM as reference.	- Reference regions of GM and WM covered a broad area in HCs and were free of voxels located in regions known to bear high tau burdens in AD and PSP. 18F-PM-PBB3 retentions in WM reference regions exhibited age-related declines. 18F-PM-PBB3 retentions for GM reference regions were unaffected by aging and provided SUVR with higher contrast than cerebellar reference regions in FTLD (CBD). Presented methodology for determining reference tissues could improve the accuracy of 18F-PM-PBB3-PET measurements of tau burden.
Aghakhanyan 2022 [94] (19)	- 24 PSP (70.9 ± 6.9 a, 12 f) - 13 non-AD MCI controls (63.8 ± 9.1 a, 4 f)	[18F]PI-2620	- Biograph mMR PET/MRI (Siemens Healthcare, Erlangen, Germany) imaging. Distribution volume ratios were estimated to assess tau burden. MRI: resting-state fMRI.	- Aberrant connectivity as shown by a significant NBS network (consisting of 89 ROIs and 118 connections) in PSP vs. controls. Significant effects of tau load on functional network connectivity in the right globus pallidus externus and left dentate nucleus. Association of the network linked with increased tau load in the right globus pallidus externus and hyperconnectivity of low-range intra-opercular connections Association of the network linked with increased tau load in the left dentate nucleus and cerebellar hyperconnectivity and corticocerebellar hypoconnectivity.
Matsuoka 2023 [101] (20)	- 20 PSP (70.7 ± 8.4 a, 11 f) - 23 HC (67.5 ± 5.1 a, 14 f)	18F-PM-PBB3	- Biograph mCT flow system (Siemens Healthcare, Erlangen, Germany) (injected dose: 185.7 ± 6.8 MBq). MRI: Single-voxel MRS.	- Correlations between apathy scale scores and 18F-PM-PBB3 SUVRs in the angular gyrus in PSP patients. Tau pathologies were observed in the subcortical and cortical structures in PSP. Glutathione levels of posterior cingulate cortex were correlated with apathy scale and tau depositions in the angular gyrus, although PSP cases did not show glutathione level alterations compared with HC. Atrophy was observed in subcortical areas, and GM volumes in the inferior frontal gyrus and anterior cingulate cortex were positively correlated with apathy scales. Synergistic contributions of tau pathologies and glutathione reductions in the posterior cortex to apathy scales, in parallel with associations of GM atrophy in the anterior cortex with apathy scales.
Oliveira Hauer 2023 [102] (21)	- 23 PSP (69.1 ± 8.4 a, 13 f) - 22 PD (70.9 ± 10.2 a, 9 f) - 25 DLB (73.9 ± 6.3 a, 5 f) - 61 HC (71.9 ± 8.8 a, 33 f)	[18F]RO948	- PET/CT scanners Discovery MI; (GE Healthcare, Waukesha, WI, USA) acquired in 4 × 5 min time frames 70–90 min after bolus injection of ~370 MBq. PET images were calculated as SUVR images with inferior cerebellar cortex region as reference.	- PET imaging with [18F]RO948 showed distinct patterns in PSP patients, with higher SUVR values in the globus pallidus and lower SUVR in the substantia nigra vs. HC, DLB, and PD. PSP patients exhibited higher SUVRs in the globus pallidus, suggesting specific changes in this brain region compared to other groups. Conversely, PSP patients had lower SUVRs in the substantia nigra, further distinguishing them from DLB and PD patients. Combining multiple biomarkers, including PET imaging, midbrain/pons area ratio (M:P), and CSF NfL levels, improved the accuracy of diagnosing and differentiating PSP from other neurodegenerative diseases. M:P ratio was also significantly lower in PSP patients vs. HC and α-synucleinopathies (DLB and PD).

## Data Availability

No new data were created in this review. Data sharing is not applicable to this article.

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
