# Peer review of "New Perspectives in Radiological and Radiopharmaceutical Hybrid Imaging in Progressive Supranuclear Palsy: A Systematic Review"

_cells, 2023, doi:10.3390/cells12242776_

Round 1

Reviewer 1 Report

Comments and Suggestions for Authors

The manuscript "New perspectives in radiological and radiopharmaceutical hybrid imaging in progressive supranuclear palsy: a systematic review" by Strobel et al is well written, well referenced and concerns an interesting topic. I have no major concerns with the paper but can make a few suggestions:

1. Authors should try to add some brief commentary as to why limitations in imaging discrimination of diseases is limited eg in some AV-1451 PET studies PD and PSP cant be differentiated, is this due to resolution, lack of tracer specificity, distribution of pathology or overlap in tau pathology between PD and PSP etc.

2. I was a bit surprised that there was no mention of MR PD index, an early MRI based volumetric approach with reasonable sensitivity/specificity for differentiating PD and PSP (Quattrone et al 2008 Radiology)

3. Authors should emphasise that at this stage, none of the available imaging modalities can reliably differentiate PSP from PD in individual patients, despite group data showing significant differences.

4. Another limitation of imaging studies is the gold standard of syndrome diagnosis in life remains clinical criteria which are not as accurate as post-mortem pathology. This means that some PSP-P misdiagnosed as PD may  enter imaging studies and affect the results. Some mention of this limitation should be made.

Reviewer 2 Report

Comments and Suggestions for Authors

Authors provide an overview on PET research papers also involving MRI in progressive supranuclear palsy (PSP). A few issues of methodological relevance are addressed only in passing. The paper is difficult to follow for non-experts as there is no systematic explanation of the underlying molecular, pathological and clinical aspects of PSP and related diseases.

1. Authors put emphasis on hybrid PET/MRI but do not address specific strengths and limitations of the hybrid technique with simultaneous scanning relative to separate PET and MR imaging with co-registration, which (as to be expected) represents the vast majority of cited studies.

2. Specifically, limitations and potential strengths of the two approaches with regards to corrections for attenuation, partial volume and movement should be presented.

3. The molecular pathology of PSP should be described in more detail for a better understanding of its relation to signal changes in PET and MRI. Also, CBD as the major differential diagnosis should be introduced clinically and pathologically.

4. The various topical subchapters show considerable overlap with respect to studies cited. It would be helpful if in the text not only citations were provided but also references to the respective entries in table 1.

5. In figure 2, arrows (or other suitable markers) illustrating target and off-target uptake areas, respectively, should be included.

6. In line 376 ff it is unclear whether structural or functional networks have been covered by MRI. Acronym NBS should be explained.

Round 2

Reviewer 2 Report

Comments and Suggestions for Authors

Appropriate changes have been made.